# [Re] Learning Intra-Batch Connections for Deep Metric Learning

1          **Reproducibility Summary**

2    **Scope of Reproducibility**

3    In this work we reproduce the paper by Seidenschwarz et al. (2021). They introduce a method that aims to more
4    successfully capture the global structure of embedding space in the task of metric learning by constructing graphs of
5    mini-batches and applying Message Passing Network (MPN) with dot product self-attention on them. They argue
6    that using MPN alleviates the need for specially designed loss functions and that their method can be trained with
7    cross-entropy loss while still achieving state-of-the-art performance on image retrieval and clustering tasks. We
8    reproduce their experiments in order to verify this main claim and further explore two additional claims.

9    **Methodology**

10   We reproduce the original experiments using slightly modified authors' code. Firstly, we reevaluate the trained models
11   provided by the authors. Secondly, we replicate their results by training our own models on CUB-200-2021, Cars196,
12   Stanford Online Products and In-Shop datasets. Furthermore, we perform additional experiments to uncover how the
13   attention mechanism evolves during training and whether that agrees with authors' secondary claims. Running all the
14   experiments took approximately 47 GPU hours on Nvidia TITAN X GPU.

15   **Results**

16   Recall@K scores reported in the paper and the results obtained by reevaluating authors' models completely agree in
17   almost every case. The same can be observed for the models we trained from scratch with differences no larger than
18   0.5$pp$ on all datasets except for Cars196. There, the discrepancies are around 1$pp$ which is also relatively low. The
19   experimental results strongly support the main claim of the paper.

20   **What was easy**

21   The authors' code was very well organized. The instructions were sufficient to create an environment and download the
22   relevant data. Once the repository was successfully set up, training and evaluating the models was very straightforward.

23   **What was difficult**

24   Initially, the code contained a few minor bugs that had to be fixed in order to run it. Because the repository is very large
25   and contains a lot of code with many possible ways to configure the models but not a lot of comments and documentation,
26   it took us more time than expected to unravel the meaning of all the possible parameters and configurations.

27   **Communication with original authors**

28   We found the original paper in combination with the authors' code sufficient to infer all the necessary information and
29   replicate the main findings. Therefore, we have not contacted the authors.

## 1   Introduction

Metric learning is a subfield in which the goal is to learn a custom, task-specific, distance metric. This is achieved by learning a mapping from the original space into some lower dimensional embedding space that pushes the similar data points closer together and different data points further apart in this new space. The mapping is often parameterized by a deep neural network. These networks are usually trained using specially designed loss functions that consider pairs or triplets of data points such as contrastive loss (Bromley et al., 1993) and triplet loss (Schultz and Joachims, 2003). Considering only pairs or triplets limits the possibility of capturing the global structure of the embedding space. Furthermore, additional tricks such as smart sampling of pairs or triplets is often required to avoid uninformative samples.

Recently, Seidenschwarz et al. (2021) proposed an approach for metric learning that addresses these drawbacks by introducing a Message Passing Network (MPN) with dot product self-attention that allows all the samples inside a mini-batch to exchange messages, thus modelling the global structure of the embedding space. This way, they move beyond considering only pairs or triplets of samples and since each sample can gather information about the other samples by message passing, there is no need to use specialized loss functions.

In this work, we reproduce their paper in order to verify its main claims. We design additional experiments that further strengthen the claims of the paper.

## 2   Scope of reproducibility

The main claims of the original paper are:

>   Claim 1: Introducing a Message Passing Network (MPN) with dot product self-attention on top of the backbone network alleviates the need for specially designed loss functions and a typical cross-entropy loss can be used instead. During inference, the MPN can be discarded and the backbone will provide representations of high quality that can achieve good results on image retrieval and clustering tasks.

>   Claim 2: Using MPN with intelligently constructed mini-batches during inference yields an additional performance improvement.

>   Claim 3: Dot product self-attention is useful since the samples inside a mini-batch attend to samples of the same class more strongly while still being able to attend to every other sample and capture the global batch structure.

The authors support claims 1 and 2 with extensive experimentation and we merely reproduce those experiments to verify their validity. Claim 3 follows from the definition of attention but our aim is to find out to what extent is the attention being utilized by the model. To support it, we introduce an additional experiment that explores the evolution of attention during training.

## 3   Methodology

We use the publicly available code published by the authors [1]. We first thoroughly verify that the implementation matches the descriptions in the paper. Next, we retrain the models on all the datasets mentioned in the paper in order to verify the validity of the first two claims. Finally, we further explore the third claim.

### 3.1   Model descriptions

We closely follow the model description from the original paper. The first part of the model is the ResNet50 (He et al., 2016) backbone pretrained on ILSVRC 2012-CLS (Russakovsky et al., 2015). The backbone produces the representation $f_i \in \mathbb{R}^d$ for every image $i$. The dimension $d$ is 512. The backbone is followed by a MPN with $L$ layers and dot product self-attention with $M$ attention heads. Each MPN layer consists of the attention block, followed by two fully connected layers with layer normalization and skip connections. In the attention block, the representation $\mathbf{h}_i^{l+1}$ of

---

[1] https://github.com/dvl-tum/intra_batch

the $i$-th node at layer $l + 1$ is defined as

$$\mathbf{h}_i^{l+1} = cat\left(\sum_{j \in N_i} \alpha_{i,j}^{l,1} \mathbf{W}^{l,1} \mathbf{h}_j^l, \ldots, \alpha_{i,j}^{l,M} \mathbf{W}^{l,M} \mathbf{h}_j^l\right),$$

where $cat$ represents concatenation, $N_i$ is the neighborhood of node $i$ and $\alpha_{i,j}^{l,m}$ is the attention score between nodes $i$ and $j$ at layer $l$ and head $m$ and is defined as

$$\alpha_{i,j}^{l,m} = \text{softmax}_j\left(\frac{(\mathbf{W}_q^{l,m}\mathbf{h}_i^l)(\mathbf{W}_k^{l,m}\mathbf{h}_j^l)^T}{\sqrt{d}}\right),$$

where $\mathbf{W}^{l,m}, \mathbf{W}_q^{l,m}, \mathbf{W}_k^{l,m} \in \mathbb{R}^{\frac{d}{M} \times d}$ are the model parameters. After the attention block, a skip connection is added and the layer normalization is applied:

$$f(h_i^{l+1}) = \text{LayerNorm}(\mathbf{h}_i^{l+1} + \mathbf{h}_i^l).$$

Then, two fully connected layers are applied and are finally followed by another skip connection and layer normalization:

$$g(\mathbf{h}_i^{l+1}) = \text{LayerNorm}(FF(f(\mathbf{h}_i^{l+1})) + f(\mathbf{h}_i^{l+1})),$$

where $FF$ denotes two fully connected layers.

During training, mini-batches are constructed such that each batch contains a sample of $n_c$ classes with $n_s$ images from each class. This results in batches of size $n_c \times n_s$. Each image in the batch is sent through the backbone and then the complete graph (each node is connected to every other node) of the batch is constructed and sent through the MPN. The final MPN representations are sent through a fully connected layer that maps them into logits. Each model is trained for 70 epochs using RAdam optimizer with learning rate $\eta$. The learning rate is divided by 10 after 30 and 50 epochs. The models are trained using cross-entropy loss and an auxiliary cross-entropy loss for the backbone. Both losses utilize label smoothing with smoothing parameter $\epsilon$ set to $0.1$ and low temperature scaling with temperature parameter $\tau$. The model is further regularized using weight decay with factor $w$.

During inference, the MPN is discarded and the backbone features are used as final representations. This alleviates the need to construct mini-batches from the test data and keeps the number of parameters similar to other metric learning approaches. Alternatively, the batches can be constructed using Reciprocal k-Nearest Neighbors, thus allowing the utilization of the whole network during inference.

## 3.2  Datasets

We use the same 4 datasets that the authors used in the original paper:

- Caltech-UCSD Birds-200-2011 (CUB-200-2011) (Wah et al., 2011) is a dataset containing 11,788 images of birds from 200 different classes. We use the first 100 classes for training and the rest for testing.
- Cars196 (Krause et al., 2013) contains 16,185 images of cars from 196 different classes. The first half of classes is used for training and the rest for testing.
- Stanford Online Products (SOP) (Song et al., 2016) consists of 120,053 product images from 22,634 different classes. The first 11,318 classes are used for training and the rest for testing.
- In-Shop Clothes (Liu et al., 2016) contains 52,712 images of clothing from 7,982 classes. The first 3,997 classes are used for training. The test set containing the rest of the classes is further split into a query and a gallery set following the annotations that come with the dataset.

During training, a random crop of the image is resized $227 \times 227$ followed by a random horizontal flip with probability 0.5. During testing, the image is resized to $256 \times 256$ and the central crop of size $227 \times 227$ is taken. In both stages, the image is finally normalized to mean and variance required by the backbone. Even though not mentioned in the paper, it is apparent from the code that the authors also use random erasing transformation (Zhong et al., 2020) during training. Therefore, we also adopt this procedure during our experiments – after the image is normalized, a random patch of area between 2% and 40% of the original image area with aspect ratio between 0.3 and 3.3 is erased (all pixels are replaced with values (0.4914, 0.4822, 0.4465)) with probability 0.5. The area and the aspect ratio are picked uniformly at random.

## 3.3 Hyperparameters

In their paper, the authors performed a random search over hyperparameter space in order to find the best configuration but the details and the chosen parameters are not clearly stated. Nonetheless, the details were easy to infer from the code. The authors performed a random search by randomly sampling 30 different configurations. Reproducing the search would mean retraining each model 30 times which would require more than 1500 GPU hours. On the other hand, the hyperparameters found by authors' random search were sufficient to closely reproduce the results reported in the paper. Therefore, we decide not to perform additional hyperparameter search and use the authors' hyperparameters instead. The hyperparameters and the search ranges considered in the original work are shown in Table 1.

Table 1: The hyperparameters used throughout the experiments. The last column indicates a range of values that the authors considered during random search. Note that the parameters are rounded for brevity. $U_c$ represents the continuous uniform distribution while $U_d$ represents the discrete uniform distribution.

|  | CUB-200-2011 | Cars196 | SOP | In-Shop | Authors considered |
|---|---|---|---|---|---|
| Learning rate $\eta$ | $1.56 \times 10^{-4}$ | $3.67 \times 10^{-4}$ | $2.47 \times 10^{-4}$ | $1.13 \times 10^{-4}$ | $10^u, u \sim U_c(-5, -3)$ |
| Weight decay $w$ | $6.06 \times 10^{-6}$ | $2.55 \times 10^{-9}$ | $2.77 \times 10^{-13}$ | $1.55 \times 10^{-7}$ | $10^u, u \sim U_c(-15, -6)$ |
| Temperature $\tau$ | 0.20 | 0.11 | 0.60 | 0.19 | $U_c(0, 1)$ |
| Number of classes $n_c$ | 6 | 10 | 15 | 14 | $U_d(6, 15)$ |
| Images per class $n_s$ | 9 | 7 | 6 | 4 | $U_d(3, 9)$ |
| MPN layers $L$ | 1 | 2 | 1 | 1 | $U_d(1, 4)$ |
| Attention heads $M$ | 2 | 8 | 8 | 4 | $2^u, u \sim U_d(0, 3)$ |

## 3.4 Experimental setup and code

The environment for running the experiments was created per instructions in the authors' repository. The data were downloaded from the links provided in the same instructions. The code initially contained a few minor bugs that prevented training the models immediately after the setup. We fixed those bugs and extended the repository with the code for additional experiments. Our code can be found at `https://anonymous.4open.science/r/reproducibility-638F`.

The models are evaluated on tasks of image retrieval and clustering using the following metrics:

- **Recall@K** – Each image in the test set is first sent through the trained network to obtain its representation in the learned embedding space. Then, $K$ nearest neighbors of the image are found using the distance metric of choice. Recall@K is a percentage of test images for which at least one of the $K$ nearest neighbors is of the same class. We use cosine distance for In-Shop and Euclidean distance for other datasets. The choice of $K$ is the same as in the original paper and coincides with the usual values reported in metric learning papers.

- **Normalized Mutual Information (NMI)** is a measure of clustering quality. The clusters are obtained using k-Means with k set to the number of classes. NMI measures the amount of information about the ground truth labels obtained by observing the labels from the clustering algorithm. It is defined as

$$\text{NMI}(G, C) = \frac{I(G, C)}{H(G) + H(C)},$$

where $G$ are the ground truth labels, $C$ are labels obtained by clustering, $I(\cdot, \cdot)$ is the mutual information $H(\cdot)$ is the entropy.

When using MPN during inference, mini-batches have to be constructed differently than during training because we have no access to ground truth labels. The authors do this by constructing reciprocal k-nearest neighbors. Consider a query image $q$. Let $N_q^k$ be the set of k-nearest neighbors of image $q$. Then $N_{r,q}^k$ is the reciprocal k-nearest neighbors set and is defined as

$$N_{r,q}^k = \{u | u \in N_q^k \wedge q \in N_u^k\},$$

i.e., out of all the nearest neighbors of $q$ we keep only those that contain $q$ in their set of k-nearest neighbors. Furthermore, we extend $N_{r,q}^k$ to $\tilde{N}_{r,q}^k$ defined as

$$\tilde{N}_{r,q}^k = N_{r,q}^k \cup \bigcup_{g \in N_{r,q}^k} \begin{cases} N_{r,g}^{\frac{1}{2}k} & \text{if } |N_{r,q}^k \cap N_{r,g}^{\frac{1}{2}k}| \geq \alpha |N_{r,g}^{\frac{1}{2}k}| \\ \emptyset & \text{otherwise.} \end{cases}$$

In other words, we extend the reciprocal k-nearest neighbor set of $q$ with the reciprocal $\frac{1}{2}k$-nearest neighbor sets of images $g$ that are in reciprocal k-nearest neighbor set of $q$ if the overlap between the two sets is sufficient. The authors set $\alpha$ to $\frac{2}{3}$. Once $\tilde{N}_{r,q}^k$ is computed, its elements are used alongside $q$ to create a mini-batch. That batch is sent through the MPN in order to refine features of image $q$.

### 3.5 Computational requirements

We run all the experiments on a server with Intel Xeon E5-2650 v4 CPU, 64 GB of RAM and Nvidia TITAN X GPU. All the training and evaluation is performed on the GPU. The training took approximately 2.5, 3.5, 10.5 and 30 hours for CUB-200-2011, Cars196, In-Shop and Stanford Online Products respectively. The evaluation took approximately 15 minutes for all the models. All in all, this resulted in approximately 47 GPU hours.

## 4 Results

This Section covers experimental results that are used to verify claims made in Section 2.

### 4.1 Results reproducing original paper

In the original paper, the authors compare their method to a plethora of different metric learning approaches. They demonstrate that their method is state-of-the-art. While reproducing results of each of the compared approaches is out of the scope of this paper, showing that our results of the authors' method agree with the ones reported in the original paper is enough to support claim 1 in Section 2. Since the authors provide pretrained models, we first load and reevaluate them. We then retrain the models with hyperparameters provided in Section 3.3 and compare both with the results claimed in the paper. The results are shown in Table 2.

Table 2: Model performance on datasets from the original paper.

|  |  | Our model | Authors' model | Paper |
|---|---|---|---|---|
| CUB-200-2011 | **R@1** | $70.1 \pm 0.60$ | $70.3 \pm 0.59$ | 70.3 |
| | **R@2** | $79.9 \pm 0.52$ | $80.3 \pm 0.52$ | 80.3 |
| | **R@4** | $87.5 \pm 0.43$ | $87.6 \pm 0.43$ | 87.6 |
| | **R@8** | $92.8 \pm 0.34$ | $92.7 \pm 0.34$ | 92.7 |
| | **NMI** | $72.6 \pm 0.34$ | $73.9 \pm 0.34$ | 74.0 |
| Cars196 | **R@1** | $87.3 \pm 0.37$ | $88.1 \pm 0.36$ | 88.1 |
| | **R@2** | $92.3 \pm 0.30$ | $92.7 \pm 0.29$ | 93.3 |
| | **R@4** | $95.1 \pm 0.24$ | $95.6 \pm 0.23$ | 96.2 |
| | **R@8** | $97.0 \pm 0.19$ | $97.5 \pm 0.17$ | 98.2 |
| | **NMI** | $70.0 \pm 0.30$ | $70.9 \pm 0.29$ | 74.8 |
| SOP | **R@1** | $81.0 \pm 0.16$ | $81.4 \pm 0.16$ | 81.4 |
| | **R@10** | $91.1 \pm 0.10$ | $91.3 \pm 0.12$ | 91.3 |
| | **R@100** | $95.5 \pm 0.09$ | $96.0 \pm 0.08$ | 95.9 |
| In-Shop | **R@1** | $92.5 \pm 0.22$ | $92.8 \pm 0.22$ | 92.8 |
| | **R@10** | $98.2 \pm 0.11$ | $98.5 \pm 0.10$ | 98.5 |
| | **R@20** | $98.8 \pm 0.09$ | $99.0 \pm 0.08$ | 99.1 |
| | **R@40** | $99.2 \pm 0.08$ | $99.2 \pm 0.07$ | 99.2 |

Recall@K scores reported in the paper and the results obtained by reevaluating authors' models completely agree in almost every case. The same can be observed for the models we trained from scratch with differences no larger than $0.5pp$ on all datasets except for Cars196. There, the discrepancies are around $1pp$ which is also relatively low. The

discrepancies between NMI scores are higher, especially for Cars196 (almost $4pp$). This could be attributed to the fact that NMI depends on the result of k-Means clustering algorithm which is known to be unstable and prone to converging to local minima.

## 4.2 Results beyond original paper

### 4.2.1 MPN during inference

The authors state the results from Section 4.1 as their main contribution. There, they use features from backbone ResNet50 to represent test images but also argue that using MPN at test time can lead to additional performance improvement. To support claim 2 from Section 2, they test their models using MPN at inference. Since ground truth labels are not known during inference, the mini-batches are constructed as described in Section 3.4. They demonstrate that this approach gives additional improvement compared to using backbone representations but they only report Recall@1 and NMI. Here, we repeat the experiment with our models and extend the results to contain all the metrics in Table 2. The results are shown in Table 3.

Table 3: Model performance with and without MPN during inference.

|  |  | Our models | | Authors' models | | |
|---|---|---|---|---|---|---|
|  |  | Without MPN | With MPN | Without MPN | With MPN | Paper |
| CUB-200-2011 | R@1 | **70.1 ± 0.60** | 69.8 ± 0.60 | 70.3 ± 0.59 | **70.7 ± 0.60** | 70.8 |
|  | R@2 | **79.9 ± 0.52** | 79.1 ± 0.53 | **80.3 ± 0.52** | 80.1 ± 0.52 | - |
|  | R@4 | **87.5 ± 0.43** | 86.3 ± 0.45 | **87.6 ± 0.43** | 86.8 ± 0.44 | - |
|  | R@8 | **92.8 ± 0.34** | 91.3 ± 0.37 | **92.7 ± 0.34** | 91.3 ± 0.37 | - |
|  | NMI | 72.6 ± 0.34 | **74.6 ± 0.35** | 73.9 ± 0.34 | **74.6 ± 0.35** | 74.5 |
| Cars196 | R@1 | **87.3 ± 0.37** | 86.9 ± 0.37 | 88.1 ± 0.36 | **88.4 ± 0.36** | 88.6 |
|  | R@2 | **92.3 ± 0.30** | 90.5 ± 0.33 | **92.7 ± 0.29** | 92.3 ± 0.30 | - |
|  | R@4 | **95.1 ± 0.24** | 93.5 ± 0.27 | **95.6 ± 0.23** | 95.0 ± 0.24 | - |
|  | R@8 | **97.0 ± 0.19** | 95.6 ± 0.23 | **97.5 ± 0.17** | 96.7 ± 0.20 | - |
|  | NMI | 70.0 ± 0.30 | **70.9 ± 0.30** | 70.9 ± 0.29 | **74.0 ± 0.29** | 76.2 |

Looking at means and standard errors, we cannot support nor oppose the claim that using MPN during inference yields a better performance.

### 4.2.2 Evolution of attention

The method introduced by the authors incorporates dot product self-attention into message passing. Theoretically, this allows each node to weigh the importance of its neighbors and to draw more information from important nodes. In practice, there exist two edge cases in which attention is practically useless:

- When attention scores are equal for every node, i.e., when they parameterize the uniform distribution over neighbors – such MPN is equivalent to the basic MPN where messages from each neighbor are just averaged.
- When each node attends only to a small number of neighbors, for example only two – in that case, this is equivalent to using handcrafted triplets and the MPN formulation decomposes into a formulation of a less powerful loss function designed for metric learning (e.g., triplet loss).

In the original paper, the authors briefly explore how attention changes across a few steps of message passing and claim that attention scores for samples of the same class get higher and higher (claim 3 in Section 2). Here, we additionally explore attention for a single node during the whole training process and explicitly check that neither of the edge cases holds. To check for the first edge case, we compute the Kullback-Leibler divergence between the attention scores of each node in a randomly chosen batch and the uniform distribution over all the nodes. The results on CUB-200-2011 are shown in Figure 1. To check for the second edge case, we inspect attention scores for a randomly chosen node in a randomly chosen mini-batch and visualize them for each training epoch. The results on CUB-200-2011 are shown in Figure 2.

It is easy to observe that none of the edge cases hold. While one attention head is very close to uniform distribution, the other diverges significantly. Similar thing can be observed by inspecting attentions for node 10. At the first head, it

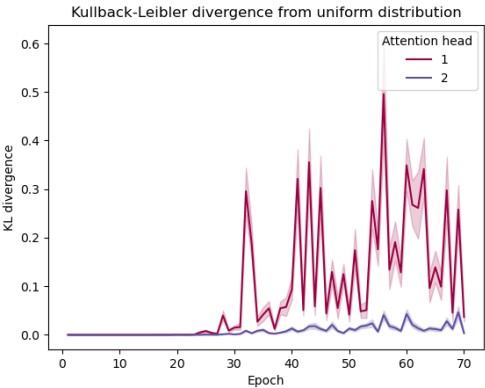

Figure 1: The Kullback-Leibler divergence from uniform distribution of attention scores for one randomly chosen mini-batch during training.

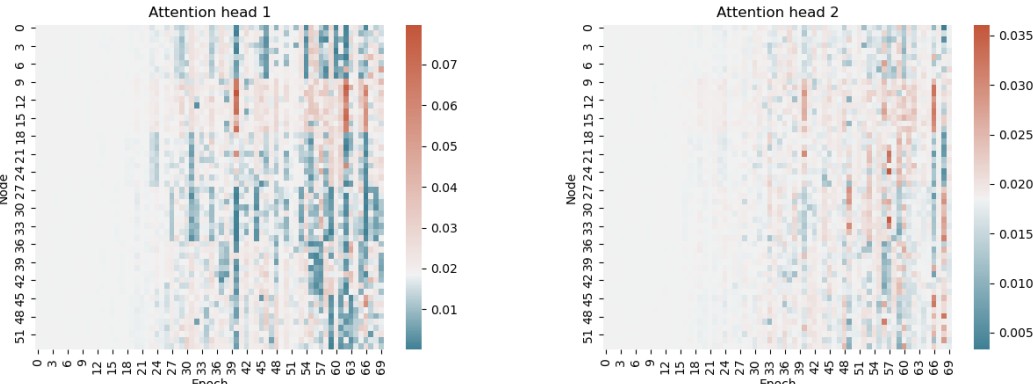

Figure 2: Attention scores for node with index 10 in a randomly chosen mini-batch. Note that every 9 consecutive nodes belong to the same class which means that node 10 is of the same class as nodes 9 – 17. Gray color represents the probability that each node would have in a uniform distribution.

attends to its neighbors of the same class much more strongly. At the second head, the attention is distributed more uniformly across all the nodes. Also, the attentions towards the same class get stronger and stronger during training (with a few exceptions). This agrees with what the authors claim. The results also confirm the fact that using multiple attention heads is useful – one head attends only to the nodes of the same class while the other head attends to other classes that might also be similar.

## 5 Discussion

The experimental results strongly support claim 1. Our retrained models agree very closely with the authors' models and the results reported in the paper with the difference of at most $1pp$ and less than $0.5pp$ for most datasets and metrics. Claim 2, which is less relevant and was only considered a bonus, can be neither supported nor opposed because the results are very similar both with and without MPN during inference. Of course, adding MPN during inference introduces a lot of overhead so being able not to use it and still achieve state-of-the-art further strengthens claim 1. Claim 3 is also supported since we clearly observed the usefulness of attention and the fact that at least one attention head focuses most of its weight on the images of the same class while the rest explore the embedding space more widely.

### 5.1 What was easy

The authors' code was very well organized. The instructions were sufficient to create an environment and download the relevant data. The repository also contains scripts that parse the datasets into required formats. Once the repository was successfully set up, training and evaluating the models was very straightforward. The paper itself is clear and contains enough information for someone to reimplement the method from scratch.

### 5.2 What was difficult

Initially, the code contained a few minor bugs that had to be fixed in order to run it. Because the repository is very large and contains a lot of code with many possible ways to configure the models but not a lot of comments and documentation, it took us more time than expected to unravel the meaning of all the possible parameters and configurations.

### 5.3 Communication with original authors

Since the paper in combination with the authors' code was sufficient to reproduce the main claims, we have not contacted the authors.

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
