# OpenReview forum: "[Re] Learning Intra-Batch Connections for Deep Metric Learning"
_ML_Reproducibility_Challenge/2021/Fall — Reject_

### Official Review · Reviewer_GPY8 · 2022-03-01

**Rating:** 8
**Confidence:** 4

**Review:**

The authors provided a mostly clear reproducibility summary, except for some missing details about how the code was slightly modified or what were the bugs in the code. In the scope of reproducibility, the two additional claims should also be included to clearly map out the rest of the paper. This is later rectified in the main text, although I would have liked to see more information on the experiments "beyond the original paper".

The methodology is clearly written and concise. Unfortunately no additional hyperparameter search was done even though the paper's code was included. The authors seemed to have no major problem reproducing the original paper. The results "beyond original paper" were especially interesting. In particular, they show that using MPN at inference is not better than not using it under all metrics. In fact, the original paper only results metrics under which it does. Moreover, the evolution of attention results present interesting visualisations that confirm the original paper's claims.

---

### Official Review · Reviewer_om1T · 2022-03-01
**Reproduction of "Learning Intra-Batch Connections for Deep Metric Learning"**

**Rating:** 5
**Confidence:** 4

**Review:**

This paper (repro-paper) reproduced the approach from the paper "Learning Intra-Batch Connections for Deep Metric Learning" (ori-paper) by leveraging the details described in the ori-paper, code and pre-trained models. The reproduction were done in two levels: In the level one, authors used the pre-trained models and reproduced all the experimental results presented in the ori-paper. In the second level, authors re-trained the model following instructions from the ori-paper. Most of the claims in the ori-papers are reproduced. The minor differences are on the results that lead to the claim 2. The re-trained model doesn't support the claim 2, but the numbers are fairly close.

The presentation of this repro-paper is decently clear. Both results and approaches are well structured. In addition to the re-production of the ori-paper's results, More experiments are showed, such as described in the Table 3 and the Figure 2. Overall, this repro-paper closely followed the ori-paper and code, with limited creative thoughts. This repro-paper is technically sound.

---

### Meta-Review · Program_Chairs · 2022-04-09

**Recommendation:** Reject
**Confidence:** 5

**Metareview:**

The reviewers note that the authors' description of the"beyond the original paper methodology" needs additional details. Consequently, the paper is not accepted.

---

### Decision · Program_Chairs · 2022-04-09

Reject